# The Influence of the Big Five Personality Traits on Residents’ Plastic Reduction Attitudes in China

**DOI:** 10.3390/ijerph20105762

**Published:** 2023-05-09

**Authors:** Yong Li, Bairong Wang, Yunyu Li

**Affiliations:** 1School of Marxism, Shanghai Maritime University, Shanghai 201306, China; liyong@shmtu.edu.cn; 2School of Economics and Management, Shanghai Maritime University, Shanghai 201306, China; 202010753008@stu.shmtu.edu.cn

**Keywords:** Conscientiousness personality, the Big Five personality traits, plastic reduction attitudes, education level, plastic crisis management, moderating effect

## Abstract

Plastic pollution has become one of the most pressing environmental issues. It is essential to understand why an individual is or is not supportive of reducing plastics. This study aims to investigate the dynamics behind residents’ plastic reduction attitudes from the lens of the Big Five personality traits. A sample of 521 residents in China was recruited and analyzed for this study. The results indicate that the Conscientiousness personality type is a reliable green personality with positive plastic reduction attitudes. Highly conscientious individuals are more responsible for the environment, and are expected to strictly follow the plastic ban policies, whereas less conscientious individuals are more likely to turn a blind eye to them. More importantly, the relationship between a Conscientiousness personality and plastic reduction attitudes is negatively moderated by education. The discovery of education’s moderating role suggests that both an inborn personality trait of Conscientiousness and post-born education can complementarily shape residents’ plastic reduction attitudes. The findings of this study deepen the understanding of the causes of pro-environmental attitudes and provide valuable insights into plastic management in China.

## 1. Introduction

Plastic pollution has become one of the most pressing environmental issues all over the world. Plastic bags, one of the most frequently used plastic products, are among the top concerns in plastic crisis management. The worldwide annual consumption of at least 5 trillion plastic bags raises an alarm about their detrimental impacts on our planet [1,2]. Plastic bags are made of non-renewable resources (e.g., petroleum) and take hundreds of years to degrade completely [3,4]. When degraded, plastics can release harmful chemicals. By consuming plants or drinking water, these chemicals can return to the human body via the food chain [5]. Additionally, meals heated in plastic bags can induce ulcers, asthma, obesity, and cancers [6], causing great harm to human health. It is also reported that over 8 million tons of plastic are found in the ocean [7], and plastic waste is also responsible for the death of millions of sea birds, turtles, and other marine creatures [8].

In this regard, the whole world is taking action to address the severe environmental problems caused by extensive plastic bag usage [9,10,11]. One effective solution to reduce people’s usage is charging fees for plastic carrier bags [9,12]. With the world’s largest population, China is representative of its plastics crisis management efforts. For example, the Chinese government introduced two national plastic ban policies in 2008 and 2020, respectively. The 2008 policies do not allow the use of plastic bags thinner than 25 µm, and only those thicker than 25 µm are permitted for use [13]. The policies also demand that all retailers charge additional fees for the plastic carrier bags used by their consumers. These policies have been proven effective, reducing the use of plastic carrier bags by 49% [12]. However, another study finds that although the Chinese government implemented the plastic ban in 2008, many people still use plastic bags in their daily lives [14]. In January 2020, the Chinese government launched another set of tougher plastic ban policies which required all supermarkets in major cities to stop using non-biodegradable plastic bags [15,16,17]. However, at present, there is an absence of studies regarding how people have responded to China’s new 2020 plastic ban policies. In view of this, this study is motivated to examine the effect of China’s new 2020 plastic ban policies.

People vary considerably in their attitudes toward environmental issues [18]. It is essential to understand why an individual is or is not supportive of reducing plastic usage. To understand how green behaviors can be encouraged, scholars have explored the psychological antecedents of pro-environmental attitudes [18,19,20,21]. One of the psychological antecedents is people’s personality traits, which have been identified as a powerful predictor in explaining people’s green attitudes [18,20,22]. For instance, Barr [23] found that altruistic people are more environmentally friendly, while competitive-oriented and selfish people are less environmentally friendly. Regarding the Big Five personality traits, Kvasova [20] found that Agreeableness, Conscientiousness, Extraversion, and Neuroticism are positively associated with green tourism in Cyprus. A study conducted in New Zealand showed that Agreeableness, Conscientiousness, and Openness are the personality traits strongly linked to environmental engagement [24]. Soutter and Mõttus [25] indicated that Agreeableness and Conscientiousness are associated with pro-environmental attitudes in the United Kingdom. However, research regarding people’s plastic reduction attitudes is limited. Hence, this study aims to narrow this research gap by examining how the Big Five personality traits modify people’s plastic reduction attitudes in China. The results of this study could contribute to the pro-environmental literature by identifying the relationship between the Big Five personality traits and environmentalism and provide constructive implications for plastic management in China.

## 2. Literature Review

Pro-environmental attitudes are defined as people’s tendency to show favor towards the environment [26]. Given the severe environmental problems caused by plastic usage, plastic reduction attitudes are representative of people’s pro-environmental attitudes. This study aims to examine the influential factors of residents’ plastic reduction attitudes in China. It has been demonstrated that education is a vital antecedent of pro-environmental attitudes [27,28]. Furthermore, existing studies have found that plastic ban awareness is a significantly influential factor of residents’ plastic reduction attitudes [2,12]. However, research regarding the influence of personality traits is limited. As the Big Five personality model has cross-cultural reliability [29], it has been widely used to study the impact of personality traits on residents’ pro-environmental attitudes [18,30,31]. In this regard, this study chooses Big Five personality traits, plastic ban awareness, and education as the investigating variables for analysis in this study.

### 2.1. Plastic Ban Awareness and Pro-Environmental Attitudes

Plastic bag waste has become a serious worldwide problem. Many countries are taking action against the extensive use of plastic bags [2,12]. For example, the Irish government introduced a product tax on plastic bags in 2002 to increase public awareness of environmental protection and change consumers’ use of plastic bags [9]. Botswana also introduced a fee for plastic carrier bags in 2007, and plastic bag usage dropped by 24% after the introduction of the policy. It has been shown that after the implementation of plastic ban policies in 2008 in China, the use of plastic bags was reduced dramatically, and the public’s awareness of environmental protection significantly improved [32]. Similarly, a plastic bag tax was implemented in 2015 in Portugal, resulting in a 74% reduction in plastic bag consumption [33]. Existing studies find that people’s awareness of plastic ban policies plays a vital role in shaping their plastic-reducing behaviors [12,34]. Attitudes always precede behaviors [35]. Thus, it is rational to posit that awareness of plastic ban policies is positively related to residents’ plastic reduction attitudes.

### 2.2. Big Five Personality Traits and Pro-Environmental Attitudes

Generally, the Big Five personality traits consist of Extraversion, Agreeableness, Conscientiousness, Neuroticism, and Openness [36,37]. Extraversion refers to an individual’s level of social, talkative, or outgoing and describes the tendency to feel confident when organizing activities and to enjoy social interactions [38]. People with high Extraversion scores may want to engage with activities regarding pro-environmental topics, such as joining environmental protection nonprofit organizations [39]. Extroverted individuals’ care for society increases their green behaviors [40]. For instance, Markowitz, et al. [41] suggest that there is a consistently positive correlation between Extraversion and pro-environmental actions. Existing studies also find a significant relationship between residents’ Extraversion personality and pro-environmental attitudes [39,42]. Thus, we posit that people with a higher score on Extraversion are more likely to hold stronger attitudes towards reducing plastic usage than those with lower scores.

Agreeableness refers to an individual’s level of altruism, empathy, compassion, and generosity [37]. Agreeable people are usually cooperative, trustworthy, and eager to help others [38,43]. It has been found that agreeable people will treat others kindly and donate more money to charity [44]. A higher level of Agreeableness is linked to greater selflessness and increased environmental concern [18]. Existing studies suggest that Agreeableness is positively related to pro-environmental attitudes and behaviors [18,22,24,39,45]. In turn, people’s pro-environmental attitudes promote their plastic reduction attitudes as a way to save our planet. Thus, it would be rational to presume that people with a higher score on Agreeableness are more likely to hold stronger attitudes towards reducing plastic usage than those with lower scores.

Conscientiousness describes the tendency of a person to be organized, responsible, and adherent to guidance and norms [38]. Conscientious people are usually concerned with the consequences of their actions and tend to plan for better future outcomes, including environmental outcomes [18,24,31]. In other words, conscientious individuals are more likely to contribute to the well-being of the whole environment. For instance, Shen et al. [46] find that Conscientiousness has a positive correlation with residents’ energy conservation behavior in Singapore. Various studies in environmental psychology have highlighted the significantly positive relationship between residents’ Conscientiousness personality and environmentalism [24,25,47,48,49,50]. Hence, it can be hypothesized that people with a higher score on Conscientiousness have stronger plastic reduction attitudes.

Neuroticism is characterized by instability, anxiety, insecurity, and anger [24,38]. Individuals who score high on Neuroticism may struggle to control their impulses and manage stress. They may respond emotionally and dramatically to minor changes [37]. One investigation reveals that Neuroticism is positively linked with eco-friendly tourist behavior [20]. Hirsh [18] finds that neurotic people demonstrate significantly high levels of environmental concern. Similarly, Soutter, Bates [39] suggest that Neuroticism is highly associated with pro-environmental attitudes. In this regard, residents’ pro-environmental attitudes could naturally translate into attitudes towards reducing plastic usage. Given the discussions above, we posit that people who score high on Neuroticism have stronger attitudes towards reducing plastic usage.

Openness describes an individual’s eagerness and desire for new ideas and the extent of a person’s imagination and creativity [38]. Open people are curious about new experiences and motivated to explore new information. Openness has consistently demonstrated a positive association with a range of pro-environmental issues, such as emissions reduction behaviors [39]. People who score higher on Openness usually have a greater concern for the environment [18,22,41,51,52] and engage in more environmentally friendly behaviors [24]. In addition, it is found that low Openness is a barrier to residents’ pro-environmental actions [39]. Pro-environmental attitudes could translate to plastic reduction attitudes. In light of this, we posit that people who score higher on Openness are more likely to have stronger attitudes toward reducing plastic usage.

### 2.3. Education and Pro-Environmental Attitudes

Education has great power to change people’s attitudes and behaviors [53] and plays a significant role in helping students become responsible citizens [54]. As for environmental issues, education could exert a positive influence by translating environmental information and knowledge into pro-environmental attitudes and behaviors. For instance, individuals with a higher level of education tend to be more environmentally friendly [28,55,56], and environmental education can promote green behavioral changes [27]. Regarding plastic reduction, it is possible that the higher an individual’s education level, the more information they have about the negative effects of plastic usage. Therefore, education could contribute to the formation of plastic reduction attitudes.

In addition to the direct impact of education on people’s pro-environmental attitudes, education has also been introduced as a moderator for predicting green attitudes [57]. In this study, it is hypothesized that education moderates the way in which the Big Five personality traits influence residents’ plastic reduction attitudes. Specifically, if an individual has a high education level, then the positive relationship between the Big Five personality traits and his/her attitudes towards reducing plastic usage will be weakened. Likewise, if an individual has a low education level, then the positive relationship between the Big Five personality traits and his/her attitudes towards reducing plastic usage will be strengthened.

## 3. Method

### 3.1. Hypothesis Design

This study aims to analyze the influential factors of residents’ plastic reduction attitudes in China from the lens of the Big Five personality traits. Based on the existing literature on the Big Five personality theory application in pro-environmental attitudes, seven variables are analyzed in this study, including the five basic variables of the Big Five personality traits, i.e., Extraversion, Agreeableness, Conscientiousness, Neuroticism, and Openness, and two additional variables, i.e., plastic ban awareness and education. To examine the influence of contextual factors, i.e., plastic ban policies, this study selects plastic ban awareness as an investigated variable. Therefore, the following hypothesis is proposed.

**Hypothesis** **1 (H1):**
*An individual’s plastic ban awareness is positively associated with plastic reduction attitudes.*


The Big Five personality traits contain the dimensions of Extraversion, Agreeableness, Conscientiousness, Neuroticism, and Openness [37]. Existing studies find a significantly positive relationship between residents’ Big Five personality traits and pro-environmental attitudes [18]. The following hypotheses are proposed.

**Hypothesis** **2 (H2).**
*An individual’s Extraversion personality trait is positively associated with plastic reduction attitudes.*


**Hypothesis** **3 (H3).**
*An individual’s Agreeableness is positively associated with plastic reduction attitudes.*


**Hypothesis** **4 (H4).**
*An individual’s Conscientiousness is positively associated with plastic reduction attitudes.*


**Hypothesis** **5 (H5).**
*An individual’s Neuroticism is positively associated with plastic reduction attitudes.*


**Hypothesis** **6 (H6).**
*An individual’s Openness is positively associated with plastic reduction attitudes.*


Education is added to the explanation model to examine how residents’ plastic reduction attitudes are influenced by educational level. Residents with higher education tend to exhibit stronger pro-environmental attitudes [28]. Additionally, education has been used as a moderator in predicting green attitudes [57]. Therefore, this study proposes the following two hypotheses.

**Hypothesis** **7 (H7).**
*An individual’s education is positively associated with plastic reduction attitudes.*


**Hypothesis** **8 (H8).**
*The relationship between the Big Five personality traits and plastic reduction attitudes is moderated by education.*


The proposed conceptual model and main variables analyzed in this study are shown in Figure 1.

### 3.2. Data Collection

We conducted a survey to learn about residents’ plastic reduction attitudes and the related influential factors in China from January to February 2021. To ensure that the survey items were clearly stated, a pilot study involving 20 individuals was carried out. Using the convenience sampling technique, participants voluntarily participated in the survey. All the investigated participants were graduated and employed residents. In total, 521 valid questionnaires were obtained.

Plastic reduction attitudes were measured by the statement, “Plastic bags should be used as fewer as possible”. Respondents choose the best option to describe their opinions (1 = strongly disagree, 5 = strongly agree).

Plastic ban awareness was measured by the question, “Have you ever heard of the 2020 plastic ban policies in China?” The respondents could choose either “yes” or “no”.

The Big Five personality traits were measured by the items adapted from Rammstedt and John [36]. For each personality trait of the Big Five model, we asked the respondents to choose which option best describes their agreement on a five-point Likert scale ranging from “strongly disagree” (1 point) to “strongly agree” (5 points). Specifically, Extraversion: (1) I see myself as someone who is not reserved; (2) I see myself as someone who is outgoing and sociable. Agreeableness: (1) I see myself as someone who is generally trusting; (2) I do not see myself as someone who tends to find faults with others. Conscientiousness: (1) I do not see myself as someone who tends to be lazy; (2) I see myself as someone who does a thorough job. Neuroticism: (1) I do not see myself as someone who is relaxed and handles stress well; (2) I see myself as someone who gets nervous easily. Openness: (1) I do not see myself as someone who has few artistic spirits; (2) I see myself as someone who has an active imagination.

For control variables, this study examined four demographic variables that had been shown to be significantly related to pro-environmental attitudes [2,24]. For instance, it was found that females were more concerned with environment than males [58]. In this study, age was categorized into 4 groups, 1 = 18–29, 2 = 30–39, 3 = 40–49, and 4 = 50 years old and above. Gender was coded as 1 for females and 0 for males. Education was measured in 4 categories, 1 = equivalent to or less than high school, 2 = undergraduate, 3 = postgraduate, and 4 = Ph.D. We also added water conservation, a typical environmentally friendly behavior, as a control variable (1 = yes; 0 = no) [59] to see if there was a spillover effect between water conservation and reducing plastic usage.

Ordinary Least Squares (OLS) regression was used to analyze the influence of the Big Five personality traits and other influential factors on residents’ plastic reduction attitudes.

## 4. Results

### 4.1. Sample Characteristics

As shown in Table 1, among the 521 respondents, 51.4% were 18–29 years old, 35.9% were 30–39 years old, 8.3% were 40–49 years old, and 4.4% were 50 years old and above. Regarding the distribution of gender, females accounted for 58.2%, and males accounted for 41.8%. As for education, the results showed that 4.8% of the respondents had education levels equivalent to or less than high school, 40.9% had undergraduate diplomas, 32.6% had undergraduate diplomas, and the remaining 21.7% had Ph.D. diplomas. Additionally, 52.4% of the respondents had the habit of water conservation in the study.

The descriptive statistics and correlation analysis of the variables are summarized in Table 2. Plastic reduction attitudes are significantly and positively related to the factors of education, water conservation, and plastic ban awareness, as well as the personality traits of Extraversion, Agreeableness, and Conscientiousness, respectively. Table 2 also shows that the mean awareness of plastic ban policies is 0.89, i.e., 89% of the respondents knew about the 2020 national plastic ban policies, suggesting a high awareness level of plastic ban policies among the investigated respondents.

### 4.2. Influential Factors of Residents’ Plastic Reduction Attitudes

The OLS regression was conducted to investigate the effects of each variable on residents’ plastic reduction attitudes. The results of the regression analysis are summarized in Table 3.

As shown in Model 4 in Table 3, plastic ban awareness is significantly and positively (β = 0.195, *p* < 0.05) associated with residents’ plastic reduction attitudes. Hence, H1 is supported. The results indicate that the more individuals are aware of plastic ban policies, the more likely they are to support reducing plastic usage. This finding suggests that plastic ban policies are effective in shaping residents’ plastic reduction attitudes in China.

As demonstrated in Model 4 in Table 3, as expected, Conscientiousness is significantly and positively (β = 0.144, *p* < 0.01) associated with residents’ plastic reduction attitudes. Thus, H4 is supported, which is in line with the results of previous studies [18,24]. However, in Model 4, the other Big Five personality traits of Extraversion, Agreeableness, Neuroticism, and Openness are not significantly related to residents’ plastic reduction attitudes. Thus, H2, H3, H5, and H6 are not verified in this study. This research finding suggests that among the Big Five personality traits, Conscientiousness accounts most for explaining residents’ attitudes towards reducing plastic usage in China.

Results regarding the influence of socio-demographic variables on residents’ plastic reduction attitudes are presented in Model 4 in Table 3. Age is not significantly related to residents’ plastic reduction attitudes. Gender exerts a significant influence (β = 0.139, *p* < 0.05) on residents’ plastic reduction attitudes. Females show stronger attitudes towards reducing plastics usage than males, which is consistent with existing findings [2,60,61]. Many studies have examined the gender difference in green attitudes and behaviors, indicating that female is a more pro-environmental gender [2,61]. A possible reason behind this finding is that, similar to a psychological barrier, the socialized perception of environmentalism as feminine reduces males’ likelihood of exhibiting environmentally friendly behaviors [62,63]. Education significantly and positively (β = 0.075, *p* < 0.05) impacts residents’ plastic reduction attitudes. Thus, H7 is supported. This finding might be attributed to the fact that individuals with higher education levels are more informed of the environmental damage caused by plastics and the importance of protecting the environment. Due to the spillover effects, the habit of water conservation is also found to be a significant predictor of plastic reduction attitudes (β = 0.155, *p* < 0.05), which is consistent with the findings of a previous study [59].

### 4.3. Moderating Effect of Education

Based on the moderating results in Model 4 in Table 3, the interaction effects are plotted in Figure 2. Overall, individuals with a higher level of education show much stronger plastic reduction attitudes. As illustrated in Figure 2, the dotted covariant line between Conscientiousness and plastic reduction attitudes is more steep under the condition of low education, while the solid covariant line is less steep under the condition of high education. This suggests a complementary interaction effect between Conscientiousness and education in shaping residents’ plastic reduction attitudes. Moreover, for individuals who score lower in Conscientiousness, education could exert a stronger influence on their plastic reduction attitudes. Therefore, H8 is supported.

## 5. Discussion

The influence of personality traits has received little attention in existing studies regarding plastic reduction. To bridge the research gap, this study examines how the Big Five personality traits influence residents’ plastic reduction attitudes in China by conducting a semi-structured survey. The major findings and implications are summarized as follows.

First, among the Big Five personality traits, only Conscientiousness significantly influences residents’ plastic reduction attitudes in China. Individuals who score higher on Conscientiousness show stronger attitudes towards reducing plastic usage, echoing the findings of existing studies [24,25,47,48,49,50]. Different from previous studies, where Extraversion, Agreeableness, Neuroticism, and Openness are found to be associated with various kinds of pro-environmental attitudes and behaviors [18,22,39,40,52], regarding reducing plastics, these four personality traits show no significant impact in this study. This finding highlights the importance of Conscientiousness in plastic management. One potential explanation is that individuals who score high in Conscientiousness tend to be responsible and, thus, are more likely to act on the benefits of the whole environment than individuals who score low. Namely, highly conscientious individuals are expected to strictly follow the plastic ban policies, whereas less conscientious individuals are more likely to turn a blind eye to the plastic ban policies. Therefore, enhancing residents’ conscientious character and sense of social responsibility is beneficial for better plastic reduction outcomes.

Second, in line with the findings of existing studies [28,55,56], education plays a crucial role in shaping residents’ pro-environmental attitudes, particularly plastic reduction attitudes, in this study. With a lower education level, respondents may have less information and knowledge about the negative effects caused by plastics and, thus, may think banning plastic is unnecessary. Hence, emphasizing environmental education and raising environmental knowledge about the plastic crisis both in schools and communities are vital for plastic management.

Third, education also serves as a moderator in the relationship between the Conscientiousness personality and residents’ plastic reduction attitudes. The discovery of the moderating effect indicates that both an innate personality trait of Conscientiousness and post-birth education can complementarily shape residents’ plastic reduction attitudes. The role of education in plastic crisis management is two-fold: first, a direct influence is observed in the improvement of people’s environmental knowledge; second, an indirect influence is found in the development of a social norm and social pressure that emphasizes environmental responsibility and plastic reduction [64]. Once this social norm and social pressure are formed, the use of plastics will be significantly reduced in the future. Meanwhile, as not all people have an innately high score in Conscientiousness, reinforcing environmental education at schools and communities becomes extremely necessary, such as including more content regarding the damage of plastics in students’ curriculum and on community bulletin boards. More green and sustainable behaviors, including recycling plastic bags and using reusable bags, could be encouraged through school education and social media publicity. Regarding enhancing community environmental education, the No Plastic Bag Campaign Day in Malaysia is a case in point [65].

Fourth, plastic ban awareness is as high as 0.89 in our study (see Table 2), and it exerts a significantly positive influence on residents’ plastic reduction attitudes. Based on this finding, we suggest that enhancing the publicity of plastic ban policies helps reduce the usage of plastics. Community-based publicity is a potential solution. For instance, the main content of the 2020 plastic ban policies can be publicized in public areas, such as supermarkets, parks, elevators, subways, and bus stations. In addition to raising plastic ban awareness, the strict implementation of the plastic ban policies is also incredibly crucial.

## 6. Conclusions

This study uses a type of pro-environmental attitude, specifically plastic reduction attitude, to clarify which of the Big Five personality traits is most firmly associated with plastic reduction attitudes and to examine the moderating role of education. First, this study finds that, among the Big Five personality traits, only Conscientiousness exhibits a significantly positive impact on plastic reduction attitudes, while Extraversion, Agreeableness, Neuroticism, and Openness do not. Conscientiousness is a significant predictor of residents’ plastic reduction attitudes. This finding suggests that conscientious individuals are more responsible for the environment, revealing the origins of green attitudes. Moreover, weak Conscientiousness may pose a barrier to reducing plastic usage. This finding necessitates future environmental protection activities to evoke and enhance people’s consciousness of responsibility. Second, education is positively associated with residents’ plastic reduction attitudes in China, and it plays a moderating role in the relationship between Conscientiousness and plastic reduction attitudes. Third, this study finds that plastic ban awareness positively contributes to shaping residents’ plastic reduction attitudes in China. Existing pro-environmental research neglects the relative importance and influence of personality traits. This study extends the application of the Big Five personality theory in the Chinese context and contributes to the current pro-environmental literature by identifying the relationship between the Big Five personality traits and plastic reduction attitudes in China. In addition, this study provides valuable implications for designing pro-environmental campaigns in the future. For example, to promote residents’ supportive attitudes towards plastic reduction, plastic crisis managers are advised to prioritize emphasizing residents’ environment protection responsibility when designing their campaigns.

## 7. Limitations and Future Research

This study has several limitations. First, this study employs self-reported measures for analysis, which makes our results vulnerable to social desirability bias. Second, using one item to measure the participants’ awareness of plastic ban policies is far from enough and deserves future research efforts to better evaluate plastic ban awareness. Third, the majority of the investigated participants are younger than 50 years old. As a result, the research findings of this study may not reflect the plastic reduction attitudes of older people. In the future, we will recruit a more representative sample.

## Figures and Tables

**Figure 1 ijerph-20-05762-f001:**
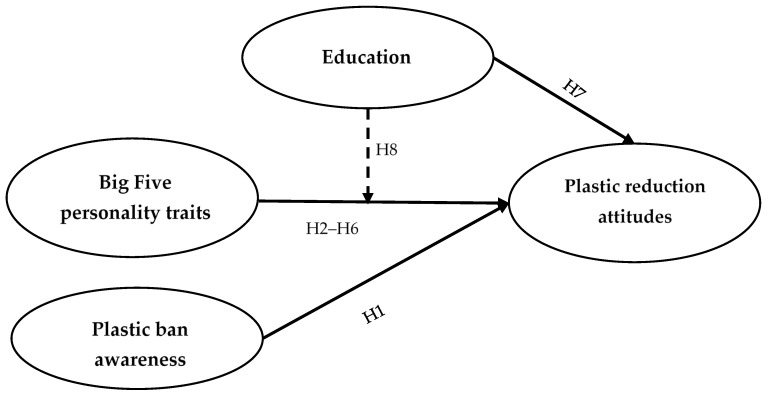
Proposed conceptual model depicting the effect of the Big Five personality traits, plastic ban awareness, and education on plastic reduction attitudes.

**Figure 2 ijerph-20-05762-f002:**
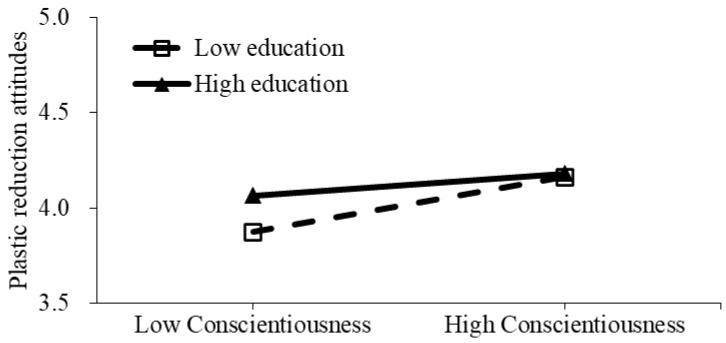
The moderating effect of education on the relationship between Conscientiousness and residents’ plastic reduction attitudes.

**Table 1 ijerph-20-05762-t001:** Summary of demographics.

Categories	Items	Ranges	Percentage (%)
Demographics	Age	1: 18–29	51.4
		2: 30–39	35.9
		3: 40–49	8.3
		4: 50 years old and above	4.4
		Total	100
	Gender	1: Female	58.2
		2: Male	41.8
		Total	100
	Education	1: Equal or less than high school	4.8
		2: Undergraduate	40.9
		3: Postgraduate	32.6
		4: PhD	21.7
		Total	100
	Water conservation	1: Yes	52.4
		0: No	47.6
		Total	100

Note: *N* = 521.

**Table 2 ijerph-20-05762-t002:** Descriptive statistics and bivariate correlation of the variables used in regression analysis.

Variables	Mean	SD	1	2	3	4	5	6	7	8	9	10	11
1. Age	1.656	0.810	1										
2. Gender	0.582	0.494	−0.168 **	1									
3. Education	2.712	0.858	0.253 **	−0.031	1								
4. Water conservation	0.524	0.500	−0.010	−0.025	−0.038	1							
5. Plastic ban awareness	0.885	0.320	−0.042	−0.038	−0.044	0.078	1						
6. Extraversion	3.398	0.909	−0.004	0.053	0.076	0.007	0.003	1					
7. Agreeableness	4.019	0.718	0.084	0.017	0.082	0.023	0.060	0.157 **	1				
8. Conscientiousness	3.501	0.772	0.116 **	−0.120 **	0.061	0.083	0.043	0.260 **	0.286 **	1			
9. Neuroticism	2.775	0.852	−0.085	0.109 *	0.037	−0.037	−0.032	−0.259 **	−0.361 **	−0.340 **	1		
10. Openness	3.624	0.843	−0.039	0.072	0.067	0.090 *	−0.043	0.297 **	0.206 **	0.203 **	−0.126 **	1	
11. Plastic reduction attitudes	4.067	0.697	0.061	0.065	0.100 *	0.125 **	0.095 *	0.093 *	0.092 *	0.173 **	−0.081	0.059	1

Notes. * *p* < 0.05; ** *p* < 0.01. *N* = 521.

**Table 3 ijerph-20-05762-t003:** Regression analysis predicts residents’ plastic reduction attitudes.

Independent Variables	Plastic Reduction Attitudes			
	Model 1	Model 2	Model 3	Model 4
Age	0.044 (0.039)	0.047 (0.039)	0.034 (0.039)	0.038 (0.039)
Gender	0.104 ^†^ (0.062)	0.110 ^†^ (0.062)	0.130 * (0.062)	0.139 * (0.062)
Education	0.076 * (0.036)	0.079 * (0.036)	0.072 * (0.036)	0.075 * (0.036)
Water conservation	0.177 ** (0.060)	0.167 ** (0.060)	0.149 * (0.060)	0.155 * (0.060)
Plastic ban awareness		0.208 * (0.095)	0.192 * (0.094)	0.195 * (0.094)
The Big Five personality traits				
Extraversion			0.028 (0.036)	0.020 (0.036)
Agreeableness			0.018 (0.046)	0.021 (0.046)
Conscientiousness			0.126 ** (0.043)	0.144 ** (0.044)
Neuroticism			−0.017 (0.040)	−0.020 (0.040)
Openness			−0.003 (0.038)	−0.003 (0.038)
Moderator				
Conscientiousness × Education				−0.061 * (0.031)
Constant	3.634 *** (0.121)	3.439 *** (0.150)	2.937 *** (0.321)	3.187 *** (0.321)
Adjusted R^2^	0.026	0.033	0.052	0.057
F test of model	4.434 **	4.543 ***	3.832 ***	3.861 ***

Notes. The table represents unstandardized regression coefficients, with standard errors in parentheses. *N* = 521. ^†^ *p* < 0.1; * *p* < 0.05; ** *p* < 0.01; *** *p* < 0.001.

## Data Availability

The data that support the findings of this study are available upon request to the corresponding author.

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
