# Peer review of "The Influence of the Big Five Personality Traits on Residents’ Plastic Reduction Attitudes in China"

_ijerph, 2023, doi:10.3390/ijerph20105762_

Round 1

Reviewer 1 Report

Respondents made a subjective assessment of their personality (personality trait), which may generate bias. Though it was mentioned in Limitations and Future Research.

What can explain that that male consumers use significantly fewer plastics?

Have similar studies been conducted in other countries?

The authors claim that reinforcing environmental education at schools becomes extremely necessary – or only in schools?

The work lacks final conclusions.  

Reviewer 2 Report

The following major revisions are proposed:

·        The title needs to be a bit more descriptive, as it leaves the topic too generic.

·        Include more relevant references in the introduction.

·        The literature review in section 2 is clearly insufficient and should be much more exhaustive and comprehensive.

·        For example, hypothesis 1 (H1) is in section 2.1 The effect of plastic ban policies, while the rest of the hypotheses are in section 2.2 Big Five personality traits and pro-environmental attitudes without a justified motive.

·        In Figure 1 the expression "Attitudes towards reducing plastic ..." is not complete.

·        Indicate for what reason this type of sampling was performed.

·        In Table 1 summary of demographic data, the male gender is missing.

·        The number of respondents is 521 and 522.

·        Specific conclusions of the study are missing.

·        The authors' contributions are not completed.

·        What appears after the bibliography should be completed or deleted.

Reviewer 3 Report

1.      This study aims to explore the human’s uncovering green personality through the conscientiousness trait pre-dicts plastic reduction attitude. The research topic is interesting and practical. However, there are some details need to modify or clarify. The suggestions are as follows.

2.      In p.2-5, there are 8 hypotheses. Although each hypothesis has the background theory, there is no whole theory framework to illustrate why the authors focus on the 8 hypotheses.

3.      In p.2-5, each hypothesis is from a big issue, but the authors did not mention why the big issue can be concentrated to be a small concept. For example, in p.2, the section 2.2 is “Big Five personality traits and pro-environmental attitudes”, and the hypothesis 2 is “An individual’s extraversion personality trait is positively associated with their attitudes towards reducing plastic usage”. The big five personality traits and pro-environmental attitudes are all big issues, but the extraversion personality trait and attitudes towards reducing plastic usage is small concept. The contents of section 2.2 can not convince readers why the big issues can be concentrated to be a small concept.

4.      Figure 1 did not show all the hypotheses. I suggest the authors add all the hypotheses in figure 1.

5.      In p.5, the authors mentioned “Awareness of plastic ban policies was measured by the question “Have you ever 208 heard of the 2020 plastic ban policies in China?” The respondents could choose either 209 “yes” or “no”.” Does it mean there just one item to confirm the participants’ awareness of plastic ban policies? Is that enough?

6.      In p.6, the percentage of age of “older than 50” is less than 5%. Will it affect the results (especially in table 2, if the participants are too less, than the correlation might distortion)? I suggest the authors provide some research limitation to explain it.

7.      In p.6, table 1, the percentage of “male” did not show.

8.      In p.8, figure 2, the authors mentioned that low and high education might be the factor to influence conscientiousness and attitudes towards reducing plastic. However, in this research, the highest percentage of age is in the period of 18-29 years old. This group of participants may just in the senior high school or university, so it is unfair to compare their education degree. It means, “the education degree” and “the age” is a factor of two interaction effects in this study. The results of this study could not attribute which factor cause the results.

9.      Although the discussion can provide some paper feedback to support the findings of this study, the statistic analysis got some logic error.

10.  The further suggestion and implication is not fruitful enough.

Round 2

Reviewer 2 Report

The following revisions are proposed:

- The different hypotheses presented are still in different sections without a justified reason.

- The conclusions should still be more detailed.

Reviewer 3 Report

The authors of this research had made a relatively complete supplementary statement for the reviewers' questions. This version of manuscript is more clear. However, the contribution and applicability of this research are still not specific enough. The authors should strengthen the contribution and application of explanation research.
